# Sustainable Competitiveness in the Case of SMEs—Opportunities Provided by Social Media in an International Comparison

**Enikő Korcsmáros [1] and Bence Csinger [2,*]**

[1] Department of Economics, Faculty of Economics and Informatics, J. Selye University, 945 01 Komarno, Slovakia

[2] Department of Management, Faculty of Economics and Informatics, J. Selye University, 945 01 Komarno, Slovakia

\* Correspondence: csinger.bence@student.ujs.sk; Tel.: +36-301987064

**Abstract:** In the extremely competitive business environment typical of the 20th century, small and medium-sized enterprises have had to face countless challenges. As time progressed, digitization and the development of information and communication technology has had increasing impacts on the lives of both individuals and businesses. Now, from an organizational point of view, social media has become a corporate strategic tool the significant role of which is indisputable. The relevance of our study can be found in the fact that social media is now one of the most popular solutions if the goal of a business is to reach a specific target audience, to receive feedback about products/services, and to initiate the immediate communication that contributes to the loyalty of consumers and customers in the long term, as well as to take advantage of cost-effective advertising opportunities. The primary goal of our research is to provide the reader with a comprehensive picture of the thinking in the SME sector regarding the corporate application of social media. In our study, following a comprehensive literature review related to the topic, we use primary data collection to examine small and medium-sized enterprises operating in Hungary and Slovakia. The reason for choosing the subjects of the research is that, taking into account the territorial size of the regions under investigation, similar districts were selected, and the regions have similar numbers of businesses engaged in economic activity. The investigation process covers two regions in the two selected countries, examining a total of 1114 enterprises. Before starting our research, two hypotheses were defined. In order to test the correctness of the hypotheses, we performed statistical analyses using the SPSS program, specifically the Mantel–Haenszel test and the chi-square test. Considering the results, the hypotheses formulated by the authors proved to be correct in the case of both countries. As a result, it can be stated that the success of traditional marketing tools used before the online space greatly influences the extent to which businesses feel their presence in social media is important.

**Keywords:** social media; SME sector; corporate challenges; international research; Hungary; Slovakia

## 1. Introduction

Today, digitization and continuous information and communication technology innovation largely determine the lives of both individuals and businesses. Their significance is unquestionable, and as a result, presence in the online space plays an increasingly strategic role in the life of organizations. Social media is one of the most important corporate assets of the 20th century. In today's knowledge-based society, the most effective means of continuous contact and information provision is the Internet, and social media within it. The timeliness of our article can be found in the fact that today, social media is one of the most popular solutions for a company to reach its target audience, receive feedback

on its products/services, and initiate immediate communication (which contributes to the loyalty of consumers and clients in the long term), all of which provides them with a cost-effective advertising opportunity. The primary goal of our study is to provide the reader with a comprehensive picture of the mindset of small and medium-sized businesses regarding the corporate use of social media. In order to achieve this goal, we carried out a comprehensive literature review for the theoretical part of our article using a wide range of domestic and foreign literature. As a result of secondary data collection, the reader gains an insight into the development path of social media in addition to the description of the challenges that arise in the lives of small and medium-sized enterprises, placing great emphasis on the presentation of the social platforms that define our everyday lives, as well as the description of the opportunities and challenges inherent in corporate social media. In the subsequent part of our article, we explain the purpose of this study, the methodology used during the research, including the data collection and data analysis methods, and then the results of the research. Regarding the practical part of the article, the authors conducted a quantitative questionnaire survey among small and medium-sized enterprises operating in two regions of Hungary and two districts of Slovakia. In the course of our research, two hypotheses were formulated, and in order to prove their correctness, the authors performed statistical analyses using the IBM SPSS Statistics 25 program. Based on the results of the research, the hypotheses formulated by the authors proved to be correct for both countries. As a result, it can be stated that the success of traditional marketing tools used before the advent of the online space greatly influences the extent to which businesses feel that their presence in social media is important. Furthermore, our results support our assumption that the corporate mindset regarding the importance of a presence in social media is significantly related to the outsourcing of activities related to online platforms, i.e., the more important its presence in social media is to an organization, the greater the willingness of corporate activities related to the online space outsourcing. In addition, we examined the effect of this way of thinking on organizations' willingness to expand their presence on online platforms, in connection with which we obtained similar results. It can be stated that the more important its presence in social media is for an organization, the more willing it is to outsource activities related to social media platforms and the more willing it is to expand its presence on social media in the future.

## 2. Literature Review

Social media has an extremely long path of development behind it, since in addition to being an integral part of the everyday life of the knowledge-based society of the 21st century, it also plays a key role in the business sphere. Social media is growing rapidly and has become indispensable to modern life [1]. We are all aware that the user base of social platforms is constantly growing and that people are spending increasing amounts of time online. Social media tools enable their users to distribute and share knowledge over the Internet [2]. As a result, in the era of the fourth industrial revolution, more and more companies are aware of the indispensability of using social media for business; today, for businesses, online presence is no longer optional but key. Social media has changed the way we interact with data and other people. In the 21st century, it is essential for many businesses to have a social media user account in order to succeed and survive amid the business competition. The relevance of this research is given by the fact that social media is a suitable channel for promoting research awareness and increasing the commitment of the target audience. Our study contributes to the finding that in addition to the fact that small and medium-sized enterprises face many challenges in the era of digitalization, it is also crucial to devote the appropriate amount of time, energy and capital to the integration of social media into company processes. The SME sector must use social networks as an essential tool for growth and be patient and aware. The road to success is bumpy and long in any case, but if the business is persistent, success will gradually reach it. The reason for choosing this research topics is that small and medium-

sized enterprises play a key role in the economy of the European Union, as they can be considered a guarantee of stability and social cohesion, and they play an important role in innovation, which is indispensable for any knowledge-based economy [3]. Thanks to the flexibility of the SME sector, it can satisfy the needs of even the most demanding customers [4]. The importance of small and medium-sized enterprises in economic life was widely recognized after the 1980s, when the focus shifted from large companies to smaller enterprises. From this period onwards, these economic units have been considered the core of development and of macro social and economic policy at the international level. In the last three decades, they have become key economic players in both developed and developing countries, especially in terms of the shift towards a more global business environment and the development of information technologies [5]. Table 1 summarizes the definition of small and medium-sized enterprises according to the standards of the European Union [6].

**Table 1.** Characteristics for the definition of small and medium-sized enterprises.

| Corporate Category | Number of Employees | Annual Net Sales OR Annual Balance Sheet Total | |
|---|---|---|---|
| Micro enterprise | <10 people | ≤2 million € | ≤2 million € |
| Small enterprise | <50 people | ≤10 million € | ≤10 million € |
| Medium enterprise | <250 people | ≤50 million € | ≤50 million € |

The SME sector plays an extremely important role in the modern economy and can be defined as the most attractive and largest innovative system today. It is a recognized fact that this group of businesses has an essential contribution to economic development [7]. Overall, it can be said that small and medium-sized enterprises have a central role in the world economy, thanks to their dynamism, flexibility, and adaptability [8], and they play a key role in a country's income generation, investments, and socially indispensable factors such as job creation and employment [9]. However, today's business environment is one of the most dynamic environments that a company faces [10]. The SME sector faces many challenges due to increased competition, adaptation to extremely fast-changing market demand, technological changes, and capacity constraints related to knowledge, innovation, and creativity [11]. Table 2 illustrates the most significant global challenges identified by Gamage et al. [12] and Zutshi et al. [13]. Peter Vitkovics [14] defined five additional factors that the small and medium-sized enterprise sector must face. In the author's opinion, the first such factor is the so-called IoT, i.e., Internet of Things, whose starting point is the fact that our world has shrunk to such an extent that by the 21st century, the continuous connection available to individuals has become the focus. As time progresses, from a corporate point of view, the issue of frugality receives increasing attention, because it is necessary to treat the Earth's resources sparingly, and if an organization ignores this, it is doomed to regress within ten years. In addition to these factors, it is also important to mention the topic of data and information flow, in connection with which the source of the problem is the increase in the lack of data and information to such an extent that people cannot absorb it. Although we have overcome a financial crisis that defined the entire world, this does not mean that a similar situation cannot occur again in the future. The SME sector must learn to always have a financing reserve and alternative sources available [14].

**Table 2.** The most significant challenges in the life of the SME sector.

| Factor | Most Significant Challenge in Keywords |
|---|---|
| The challenge of global economic competition | • Low technological base, low productivity<br>• High concentration of labour-intensive sector<br>• Low entry barriers, lack of new knowledge<br>• Relatively low, constant production costs<br>• Trade liberalization<br>• Inadequate budget [15]<br>• Access to credit [16]<br>• Inefficient marketing [17] |
| Global capital and the challenge of the economic crisis | • Dependence on customers, suppliers and markets<br>• Declining pace of development, increase in the number of bankruptcies<br>• Daily crisis management [18] |
| Challenges caused by ICT | • Poor quality infrastructure network<br>• Late adaptation, dependence on risk partners<br>• Scarcity of ICT skills, lack of financial resources |
| Impact of multinational enterprises | • Influencing product competition |
| The impact of transnational enterprises | • Local procurement mechanisms<br>• Lack of knowledge transfer |
| International terrorism and religious conflicts | • Value chain disruption<br>• Adverse effect on industrial development<br>• Labor migration<br>• Risk increase |
| International trade war | • The chance of ending global economic expansion<br>• Slowing down of the global economic crisis [19] |
| International dumping | • Price discrimination<br>• Victims of export dumping [20] |
| Interruption of activity caused by an epidemic situation | • Expectation of expertise, flexibility and perseverance |
| Existential challenges | • COVID-19—corporate operations, income, trust |

In addition to an overview of the global challenges, it is important to examine which are the most significant trials that affect the European small and medium-sized enterprise sector. In their 2019 report, Michelle Alessandrini and her colleagues [21] formulated three factors. The first aspect worth mentioning is the limited availability of skilled labor. At the European Union level, small and medium-sized enterprises often experience problems in hiring skilled workers due to competition from large companies and general financial conditions. EU SMEs do not invest enough time, energy, and capital in training existing employees and are often held back by increased labor and individual costs. The second factor includes difficulties in accessing financing, which has been one of the most important issues for small and medium-sized enterprises since the economic crisis in 2008. The financial needs of the SME sector are still significant, especially with regard to fixed investments and working capital. It is also important to mention that access to state financial support is still limited for small and medium-sized enterprises. As a third and final challenge, the authors deal with the issue of excessive regulation and administrative burden. They say that in the case of the small and medium-sized enterprise sector, the compliance costs and requirements have a greater influence on their innovation capacity, the trained human capital at their disposal and hinder their internationalization [21]. After our global

and European study, we will examine what are the challenges typical of small and medium-sized enterprises at the local level. These factors are illustrated in Table 3 [13,22,23].

**Table 3.** Challenges affecting small and medium-sized enterprises at the local level.

| Author(s) | Year | A Challenging Factor |
|---|---|---|
| Sandiso Ngcobo Sandheel Sukdeo | 2015 | • Government policies and laws<br>• Lack of adequate information<br>• Financing difficulties<br>• Extremely competitive business environment<br>• Dominance of large corporations<br>• Inaccessible markets<br>• Lack of self-confidence |
| Peter Holicza | 2016 | • Lack of preparation for starting a company<br>• Insufficient financial contribution<br>• Serious loss in case of failure<br>• Complicated administrative processes |
| Ambika Zutshi et al. | 2021 | • Resource pooling for fixed operating costs and variable costs<br>• Existential questions due to the failure to cover costs<br>• Experiencing a liquidity crisis due to the lack or issue of reimbursement of permanent operating costs<br>• Pressure on production processes due to the lack of human resources and the increase in the ratio of human costs |

Overall, it can be said that the small and medium-sized business sector must carry out innovative activities in order to maintain their competitiveness and continue to develop in a dynamic and competitive environment [10]. In order to lay the foundation for exploring the opportunities and challenges that corporate social media provides for various communication-based processes in the organizational environment, it is necessary to accept the following definition of corporate social media: **"***Organizational social media includes web-based platforms that which enable employees to communicate with colleagues, broadcast messages within the organization, identify or tacitly disclose communication partners, publish or edit texts and files referring to themselves or others, communicated, published, edited by anyone within the organization, viewing organized messages, contacts, text files at a selected time***"** [24] (p. 764). The importance of social media in business life is growing at an extremely fast pace [25]. The business environment is changing very quickly with the rapid adoption of social media and the development of mobile technologies [26]. Social media platforms enable interactive experiences between brands and consumers [27]. As more businesses feel the need to join social media platforms, the social media industry is expected to continue to show a growing trend. As a result, businesses must take advantage of the opportunities offered by an online presence, but they must also be aware of the challenges of organizational use of social media [25]. Social media plays an extremely important role in business life and contains many opportunities. Businesses now have to take advantage of the opportunities offered by the online space. As a first step, they need to exploit the social media channels relevant to them in the best possible way [25]. Today, consumers are given a new role in social media. Consumers are becoming **"content creators"** and thus functional consumers, thanks to which the way of consumption is changing. Social media applications including blogs, microblogs, podcasts, and video and photo sharing sites have greatly facilitated this change. In light of this, the corporate integration of social media is extremely useful [28]. Svatošová [29] stated that social media is not just a communication tool. In his opinion, social media offers many opportunities to use promotional tools. Two years later, Svatošová's statement was confirmed by Polanska [30], who stated that one of the most obvious uses of social media in business is its use in promotional activities. With

the help of social media, a business can quickly and efficiently contact its target audience, who provide real-time, direct feedback to the organization, providing an opportunity for the continuous development of programs and analyses for measuring efficiency [29]. Considering the business side, the two main benefits of social media are cost reduction and revenue growth. Achieving these benefits is facilitated by the digital world, by providing the opportunity to share information and skills, to track consumers, to provide assistance, and to engage the target audience [31]. Today, we live in a world where information is a key resource. As a result of this, in the 21st century, businesses must take advantage of the opportunity to connect with their target audiences by actively using the previously mentioned social media platforms [32]. The online space has many features that allow you to create much richer publication channels than traditional media. It is important to mention that social media not only promotes the dissemination of information but also gives businesses the opportunity to disclose information processing. In addition, the online space enables two-way communication between individuals, consumers and businesses, giving the opportunity to understand and continuously monitor the needs of market participants, which is a key factor in a world where needs are constantly changing [33]. For instance, 71% of people are more likely to recommend a brand that has positive feedback on social media [25]. In 2017, Brunner concluded that social media is an effective marketing tool for businesses in the 21st century that greatly influences consumer purchasing decisions. Although organizations increase their advertising costs, they quickly recover the success achieved through online marketing activities. Most often, people make their purchase decisions based on social networking sites, which is extremely useful for companies to discover new, innovative ideas and to meet constantly changing consumer needs [34]. In the age of the digital revolution, the aim of businesses is to continuously monitor and share knowledge about the market, their consumers and their own employees, for which social media platforms provide an excellent opportunity. This kind of knowledge provides an opportunity to manage the involvement of stakeholders, which contributes to improving the company's reputation [35]. In addition to the advantages mentioned so far, social media also plays a role in promoting the brand, facilitating communication, increasing sales, sharing information in a business context, and providing social support to consumers. It should also not be overlooked that from a corporate point of view, the networking of consumers through social media contributes to the creation of shared value, which also affects trust in the long term [36]. Summarizing the advantages of a corporate presence in the online space, Venkateswaran et al. [37] defined the advantages of social media in business organizations in 12 points, which are summarized in Table 4.

**Table 4.** The benefits of integrating social media into your company.

| Advantages | Characteristics |
|---|---|
| Audience recognition | Tracking people's activities |
| Reaching an audience effectively | Reach people with similar interests |
| Cost effectiveness | Low cost compared to traditional media |
| Immediate feedback | Quick, honest, detailed feedback |
| Personalized service | Deeper contact and personalized customer service |
| Creating a corporate identity | Building credibility, brand, image and reputation |
| Improving market intelligence | Track the competitive landscape, providing vital information and statistics in the industry |
| Accelerating innovation and product development | Facilitating the exchange of knowledge and expertise, accelerating innovation and the development of new products |
| Increase visibility | Increase website traffic |
| Facilitating labor recruitment and recruitment | An effective tool for recruiting new talent |
| Building trust and loyalty | Creating consumer trust and loyalty |

In addition to the many advantages of social media, it is important to examine the challenging factors that businesses face in terms of presence on each platform [38]. Despite the vital role social media plays in communication, collaboration, education, etc., it also involves many challenges [39]. In 2012, Svatošová formulated six components that businesses must face in connection with their presence on online platforms. According to Svatošová [29], in the digital space, in addition to the fact that not all target groups can be reached and organizations have to deal with a large number of competitors, interactivity and immediate response often carry the risk of negative reactions and the spread of rumors. Furthermore, the author highlights that the use of social media platforms can lead to the misuse of personal data and contacts; the special software required to monitor and evaluate marketing campaigns is expensive; and due to the rapid saturation of advertisements appearing on social media sites, the information to be conveyed may be ignored, and the risk is high [29]. As mentioned earlier, social media is developing at an extremely fast pace, and this is a serious challenge: Since the platforms are constantly changing, more and more websites are appearing, which is difficult to follow. It is also a serious challenge for organizations to be aware of exactly what information needs to be monitored and developed in order to be able to perform analyses for the sake of efficiency. From a financial point of view, although a presence on social platforms is a cost-effective solution, sales do not always follow consumer needs. We must not forget about certain ethical limitations either, since improperly influencing brand communication often leads to a loss of credibility among consumers. In addition to achieving commercial goals, it is also important to place great emphasis on creating social value. The majority of consumers are completely unaware that businesses are monitoring them, which raises ethical questions. In addition, the misinterpretation of monitoring data is also a critical factor [38]. In his 2016 research, Todor concluded that presence in the digital space gives competitors the opportunity to copy marketing campaigns and use logos to deceive consumers [40]. Tuten and Min-tu-Wimsatt [41] stated that with the help of social media, consumers have the opportunity to write fake reviews (either positive or negative feedback) and to endorse different products/services for profit without actually using them. In addition to summarizing the benefits of social media, Venkateswaran and his colleagues also collected the factors that pose challenges for businesses; according to the authors, five key factors make it difficult for organizations to be present online, which are listed in Table 5 [37].

**Table 5.** Disadvantages of corporate social media integration.

| Disadvantages |
| --- |
| 1　Updating social media accounts takes time and energy. |
| 2　Social media is not completely free. |
| 3　The information is only visible for a short time before it is replaced by new posts. |
| 4　Declining personal communication. |
| 5　Loss of control over social media marketing processes. |

Overall, it can be said that despite the fact that there are many positive factors influencing social media in business life, organizations should always examine the disadvantages and risks caused by an online presence in order to achieve and maintain efficiency in the long term [37]. With the right solutions, challenges can become opportunities that provide organizations with a competitive advantage [38].

## 3. Materials and Methods

Many researchers have proven the advantages of corporate integration of social media, as well as the organizational potential of online presence. However, an overwhelming number of businesses today remain unsure and skeptical about the power of social media and its successful use. The aim of our research is to find out the opinions of businesses about their ways of thinking in relation to social media by applying a quantitative research

method. With our research results, we hope to help those businesses that are still unsure about the potential of using social media. In order to achieve our goals, we used a quantitative research method among the SME sector operating in Hungary and Slovakia. During the selection of the research subjects, we took into account the characteristics formulated by Machová-Véghová [3], according to which the SME sector plays a key role in the economy of the European Union, since society considers these enterprises to be the guarantee of stability and social cohesion, and they play an indispensable role in innovation, which can be considered one of the cornerstones of social media. In terms of the execution of the investigation, we administered our questionnaire twice. The first query was carried out in 2020 in two regions of Hungary; this research targeted the SME sector of the Central Transdanubian and Western Transdanubian regions. The second query took place in 2021, during which small and medium-sized enterprises operating in two districts of Slovakia were the focus of the research. In the case of Slovakia, the authors examined the regions of Nitra and Nagyszombat. The regions and districts were chosen because of their similar sizes and because there were similar numbers of enterprises conducting economic activities in each. In both cases, the research tool contained the same 27 questions, which were structured based on the groups of questions in Table 6.

**Table 6.** Construction of study question groups.

| Question Group 1 | Question Group 2 | Question Group 3 | Question Group 4 | Question Group 5 |
|---|---|---|---|---|
| Demographic data | Information about social media presence | Manage tracking and feedback | Expanding your social media presence and how to do it | Application and success of traditional marketing tools |

After a test phase, the questionnaire was sent to the research subjects in the form of direct mail. The questionnaire was available to all businesses, and it was completed online, ensuring complete anonymity for the respondents. In the case of both countries, approximately 20% of the sent questionnaires were returned. Specifically, 591 responses from Hungary and 523 responses from Slovakia were received that could be evaluated and used for the analysis. After completing the survey and receiving the answers, the data were organized, cleaned and coded for the sake of transparency, which processes were implemented with the help of the IBM SPSS Statistics 25 statistical program. SPSS Statistics 25 was also used to evaluate the systematized, cleaned and coded data. Before starting the research process, the authors formulated two hypotheses, which are illustrated in Table 7.

**Table 7.** List of formulated hypotheses.

| | Hypotheses |
|---|---|
| H1 | The success of previously used marketing tools has a significant relationship with the way of thinking of businesses regarding the significance of their presence in social media. |
| H2 | Corporate thinking about the importance of social media presence is significantly related to the outsourcing of social-media-related activities. |

The starting point of the study is the success of previous marketing tools, which according to the authors' assumption has a significant relationship with the importance of presence in social media (H1). The authors also assume that the importance of presence in social media is related to the outsourcing of activities related to the digital space (H2). In examining the correctness of our hypotheses, we used several statistical methods, focusing on the Mantel–Haenszel test and the chi-square test. During the data analysis, it is crucial to identify the measurement levels of the individual variables because this information is essential for the application of the statistical program, and the analysis method can be determined with the help of the measurement level of the variables [42]. In this

study, of the four variables selected by the authors, two were ordinal and two were nominal. During the analyses, a significance level of 5% was determined. The two variables defined during the first hypothesis of the research were ordinal variables measured on a 5-point Likert scale: The authors performed a Mantel–Haenszel trend test included in the chi-square test statistics in order to reveal the linear relationship between the variables. The test can be calculated as follows:

$$M^2 = (n − 1)r^2 \tag{1}$$

where n is the size of the sample (the total number of people responding to both row and column variables) and r is the sample estimate of the correlation from the given scores. During the examination of the following hypothesis, the authors used a cross-tabulation analysis, focusing on the chi-square test. Cross-tab analysis is a widespread statistical analysis method that examines the relationship between two or more variables and shows the combined frequency distribution of these variables. The creation of the chi-square test is attributed to the English mathematical statistician Karl Pearson, who created the foundations of modern statistics [42]. The formula for the Chi-square test is as follows:

$$X^2 = \sum (O_i − E_i)^2/E_i \tag{2}$$

where O is the observed frequency and E is the expected frequency. During the research, the authors also used Cramer's V index to examine the closeness of the relationship between the two variables. Cramer's V association measure is based on the Pearson Chi-square statistic and was published by Harald Cramer in 1946 [42]. The formula of the coefficient can be determined as follows:

$$V = \sqrt{X^2/N(k − 1)} \tag{3}$$

## 4. Results

In the course of our research, we conducted investigations in two regions each in Hungary and Slovakia. In the case of Hungary, the subjects of the research were small and medium-sized enterprises operating in the Western Transdanubian and Central Transdanubian regions; in the case of Slovakia, after the Hungarian research carried out in 2020, the authors in 2021 extended their investigation to the SME sector in the Nyitra and Nagyszombat regions.

*Hypothesis Analysis*

Two hypotheses related to the research topic were formulated by the authors. Today, technology is constantly changing, and when a brand becomes part of social networks, it is obvious that it can change with it. Companies around the world are increasingly using social media to tap into emerging market opportunities. For many years, however, businesses used traditional advertising channels. Marketers have used traditional forms of marketing to motivate their potential customers to continue buying and using their products/services. However, the concept of social media marketing is now at the top of many business leaders' agendas. Today, in the 21st century, people do not find time to communicate with each other in person. On the one hand, social media helps in this, but on the other hand, with its rise, traditional media experienced a decline both in business life and in their popularity [43]. Taking into account the first hypothesis, the authors assumed a significant relationship between the success of the marketing tools previously used by businesses and the ways of thinking related to the importance of the presence of organizations in social media. In the case of the two selected variables, the subjects of the research had the opportunity to answer the authors' questions on a five-point Likert scale, as a result of which it can be concluded that we can speak of two ordinal variables; the Mantel–Haenszel test provides an excellent opportunity to measure them. As the first step in conducting the study, the data measured on the Likert scale received were imported,

after which the authors made the necessary settings, the variables were renamed, and the scale data were entered for the values. As a next step, the authors ran and evaluated the test, first for Hungary and then for Slovakia. Tables 8–10 illustrate the cross-tabulation for Hungary and the results of the related chi-square test and correlations.

**Table 8.** The cross table related to hypothesis 1—Hungary.

| | | The importance of social media presence | | | | | |
|---|---|---|---|---|---|---|---|
| | | Not at all | A little bit | Can not decide | To a great extent | Fully | Altogether |
| Success of previous marketing tools | Not at all | 3 | 2 | 4 | 5 | 3 | 17 |
| | A little bit | 3 | 4 | 9 | 9 | 11 | 36 |
| | Cannot decide | 7 | 10 | 48 | 51 | 52 | 168 |
| | To a great extent | 7 | 5 | 31 | 45 | 61 | 149 |
| | Fully | 5 | 6 | 16 | 12 | 38 | 77 |
| | Altogether | 25 | 27 | 108 | 122 | 165 | 447 |

Looking at Table 8, it is clear that in the case of Hungary, taking into account the two variables, a total of 447 evaluable responses were received from businesses. Furthermore, the table illustrates the distribution of the responses received in terms of the two examined variables. The following table describes the results obtained from the chi-square test.

**Table 9.** The chi-square test related to the 1st hypothesis—Hungary.

| | Value | df | Asymptotic Significance (2-Sided) |
|---|---|---|---|
| Pearson Chi-Square | 25.493[a] | 16 | 0.062 |
| Likelihood Ratio | 24.533 | 16 | 0.078 |
| Linear-by-Linear Association | 8.253 | 1 | 0.004 |
| N of Valid Cases | 447 | | |

[1] [a] 8 cells (32.0%) have expexted count less than 5. The minimum expected count is .95

Table 9 presents the Pearson's chi-square test result, which in this case is 25.494, as well as the linear-by-linear association representing the result of the test, the value of which is 0.004 in the case of the first hypothesis. Table 10 presents the Pearson's correlation. For the successful evaluation of the test, all three tables play an important role, since thanks to the cross table, we obtain a comprehensive picture of the descriptive statistics data.

**Table 10.** The Pearson's correlations related to hypothesis 1—Hungary.

| | | The Importance of Social Media Presence | Success of Previous Marketing Tools |
|---|---|---|---|
| The importance of social media presence | Pearson Correlation | 1 | 0.136 ** |
| | Sig. (2-sided) | | 0.004 |
| | N | 447 | 447 |
| Success of previous marketing tools | Pearson Correlation | 0.136 ** | 1 |
| | Sig. (2-sided) | 0.004 | |
| | N | 447 | 447 |

[2] **Correlations is significant at the 0.01 level (2-tailed).

The table showing the result of the chi-square test is indispensable because of the Mantel–Haenszel value, and the correlation table provides key information in determining the Pearson's r coefficient (Table 10). In this case, the coefficient shows a value of 0.136. In the case of Hungary, the Mantel–Haenszel test shows a statistically significant linear association, as $X^2 = 25.493$, $p < 0.005$, r = 0.136. As a result, the null hypothesis is rejected, and the alternative hypothesis is accepted. As a next step, we examine the case of Slovakia. Table 11 illustrates the cross-tabulation related to hypothesis 1 in relation to Slovakia.

**Table 11.** Crosstab related to Hypothesis 1—Slovakia.

| Success of previous marketing tools | The importance of social media presence | | | | | |
|---|---|---|---|---|---|---|
| | Not at all | A little bit | Can not decide | To a great extent | Fully | Altogether |
| Not at all | 6 | 5 | 3 | 3 | 1 | 17 |
| A little bit | 7 | 8 | 23 | 13 | 14 | 36 |
| Cannot decide | 12 | 10 | 46 | 52 | 88 | 168 |
| To a great extent | 2 | 5 | 18 | 34 | 27 | 149 |
| Fully | 6 | 3 | 6 | 1 | 9 | 77 |
| Altogether | 33 | 31 | 96 | 103 | 139 | 402 |

The table reflects that a total of 402 evaluable responses were received from small and medium-sized enterprises. As the next step of the test, we performed the chi-square test, the results of which are presented in Table 11.

In the case of Slovakia, the Pearson's chi-square is 70.017, and the linear-by-linear association symbolizing the test result is 0.003. As the last step of the Mantel–Haenszel test, the Pearson's r coefficient was also determined for Slovakia, which is presented in Table 12.

**Table 12.** Chi-square test for hypothesis 1—Slovakia.

| | Value | df | Asymptotic Significance (2-Sided) |
|---|---|---|---|
| Pearson Chi-Square | 70.017 [a] | 16 | 0.000 |
| Likelihood Ratio | 63.152 | 16 | 0.000 |
| Linear-by-Linear Association | 9.129 | 1 | 0.003 |
| N of Valid Cases | 402 | | |

[3] [a] 6 cells (24.0%) have expected count less than 5. The minimum expected count is 1.39

Similar to Hungary, we can speak of a positive linear association in the case of Slovakia, as the resulting value of 0.151 is positive, and the relationship was also statistically significant in that $X^2 = 70.017$, $p < 0.005$, r = 0.151. In summary, the authors performed a Mantel–Haenszel test for the first hypothesis in order to examine whether a linear association can be found between the success of previously used marketing tools and the importance of social media presence. Both variables examined in the first hypothesis were ordinal variables measured on a 5-point Likert scale, where a value of 1 represented not at all, while a value of 5 represented complete agreement. Taking into account the results obtained during the investigation, it can be said that in both cases the null hypothesis was rejected and the alternative hypothesis was accepted, as a result of which it can be stated that in the case of both examined countries, a linear association could be found between the success of previously used marketing tools and the importance of presence in social media. The obtained result allows us to conclude that the two variables examined during the first hypothesis have an influencing effect on each other. Over the past decade, re-

searchers have argued that all but a few key traditional marketing tools, especially promotional mix categories (such as advertising, promotion, public relations, campaigns and consumer services), should be integrated into social media platforms. Salesforce surveyed marketers and found that marketers' mindsets about social media ROI completely changed over the course of a year, so much so that they declared that this type of marketing channel has a direct impact that can also affect the sales performance of the country [44]. Begiqiri-Bello [44] and Venkateswaran et al. [37] established that businesses today use more social media marketing compared with traditional marketing for the following advantages: time, target audience, cooperation, and cost. In the case of the second hypothesis, the authors hypothesized a relationship between a company's mindset regarding the importance of a social media presence and its willingness to expand its social media presence. Table 13 presents the results obtained from the testing of the second hypothesis in relation to Hungary and Slovakia.

**Table 13.** Pearson correlation related to hypothesis 1—Slovakia.

|  |  | The Importance of Social Media Presence | Success of Previous Marketing Tools |
|---|---|---|---|
| The importance of social media presence | Pearson Correlation | 1 | 0.151 ** |
|  | Sig. (2-sided) |  | 0.002 |
|  | N | 402 | 402 |
| Success of previous marketing tools | Pearson Correlation | 0.151 ** | 1 |
|  | Sig. (2-sided) | 0.002 |  |
|  | N | 402 | 402 |

[4] **Correlations is significant at the 0.01 level (2-tailed).

Table 14 provides a comprehensive picture of the results of the Chi-square test related to hypothesis 2 in relation to both Hungary and Slovakia.

**Table 14.** Chi-square test related to hypothesis 2

| Hungary | | | |
|---|---|---|---|
|  | Value | df | Asymptotic Significance (2-sided) |
| Pearson Chi-Square | 32.364[a] | 12 | 0.001 |
| Likelihood Ratio | 36.630 | 12 | 0.000 |
| N of Valid Cases | 448 |  |  |
| Slovakia | | | |
|  | Value | df | Asymptotic Significance (2-sided) |
| Pearson Chi-Square | 49.619[a] | 12 | 0.000 |
| Likelihood Ratio | 61.954 | 12 | 0.000 |
| N of Valid Cases | 402 |  |  |

[5] [a (Hungary)] 8 cells (40.0%) have expected count less than 5. The minimum expected count is 1.04.
[6] [a (Slovakia)] 6 cells (30.0%) have expected count less than 5. The minimum expected count is 3.01.

In response to the second hypothesis, the number of evaluable responses returned from enterprises in Hungary was 448, and in the case of Slovakia, it was 402. The reason for this can be found in the fact that only those organizations who, in connection with our previous question, declared that they were present on social media had the opportunity to answer the question. The Hungarian and Slovak results obtained during the study are also lower than the specified 5% significance level, as a result of which it can be stated that the two variables under study have a significant relationship with each other for both countries, i.e., the null hypothesis must be rejected in this case, and the alternative and

hypothesis is accepted. Based on the results, it can be stated that the more important a company's presence in social media is, the more willing it is to outsource activities related to its online presence. During the analysis, the authors also used a symmetric indicator to determine the strength of the relationship. Since a nominal and an ordinal variable were examined in the case of the second hypothesis, Cramer's V coefficient was used to describe the closeness of the relationships. The obtained results are illustrated in Table 15.

**Table 15.** Symmetric indicator related to hypothesis 2

| | | Hungary | |
|---|---|---|---|
| | | Value | Approximate Significance |
| Nominal by Nominal | Cramer's V | 0.155 | 0.001 |
| N of Valid Cases | | 448 | |
| | | Slovakia | |
| | | Value | Approximate Significance |
| Nominal by Nominal | Cramer's V | 0.203 | 0.000 |
| N of Valid Cases | | 402 | |

With the help of the indicator, we established that in the case of the second hypothesis, there was a medium to weak relationship in the cases of both Slovakia and Hungary. The results of the tests for both hypotheses show a significant relationship for both countries. Employees play a key role in digital marketing, as they implement the company's strategy. In relation to this, contrary to our research results, during the research conducted by Tiago et al. [45], feedback was received from marketing directors that digital marketing processes should not be outsourced, as they prove to be strategic tasks in the life of a business. In contrast, and in line with our research results, Pandya [46] collected ten advantages of outsourcing digital marketing activities in 2017, listed in Table 16.

**Table 16.** Benefits of outsourcing the digital marketing activities.

| | Benefits |
|---|---|
| 1 | Building an optimized digital marketing strategy with the involvement of experts |
| 2 | Delivering a new image of marketing efforts |
| 3 | Free in-house resources to focus on core business requirements |
| 4 | Application of customized solutions |
| 5 | Access through appropriate channels |
| 6 | Cost savings |
| 7 | Increase marketing efforts |
| 8 | Reduction of general costs |
| 9 | Get the best in the industry |
| 10 | Applying a proven process for launching a campaign |

## 5. Discussion and Conclusions

In the course of our investigation, in the case of both countries, we came to the conclusion that a little higher than 76% of the surveyed businesses are present in social media, which result can be interpreted based on two approaches. Based on the first approach, we concluded that the online world is an integral part of most of our respondents. However, even though our lives today would be almost unimaginable without the Internet and social media, 24% of businesses were still not present on these digital platforms. According to the authors, the primary reason for this is that as time progresses, more social media platforms are being created, and existing websites are constantly changing, with which it is extremely time-consuming to keep up. Businesses often ask questions such as: Which

social media platform do we use? How do we start? In addition, the thinking that it is not necessary to devote resources to a presence in social media often causes problems since the results are difficult to measure. However, the opportunity is given, and the strategy is available and often cost-effective for those who want to plan their presence in social media. The mindset that social media-related results are unquantifiable is no longer valid.

In addition, the authors feel that the statements made during our previous research are correct, according to which the reason for the lack of presence in social media is often to be found in the fact that businesses are afraid to invest time, energy and capital in the application of methods that represent uncertainty for them and cause change, and in many cases they are not aware of the opportunities provided by social media. Although Facebook proved to be the most popular social media platform from a corporate point of view in both countries, the authors recommend that the SME sector establish their presence on the social media platforms used by their own target audiences rather than the websites preferred by larger groups of consumers. In terms of what businesses use each social media platform for, different answers were received from the two countries under investigation. Based on the authors' proposal, businesses should focus on providing information, as well as building and increasing brand awareness, which can greatly contribute to, among other things, revenue growth, creating a competitive advantage, and increasing the number of target audiences and the possibility of breaking into new markets. Regarding outsourcing activities related to social media, similar results were obtained in both Hungary and Slovakia, as less than a third of the respondents employed external person/persons/enterprises to perform these tasks, which, according to the authors, can be attributed primarily to economic reasons. As a result of this, it is recommended that small and medium-sized enterprises place greater emphasis on the appropriate training of employees, which, although a more time-consuming process, can provide a cost-effective solution in the long term. Continuing the investigation, the authors came to the conclusion that participation in social media is of above-average importance in the lives of businesses operating in the examined countries. In this regard, the authors recommend them not to be afraid to invest time, energy and capital for a quality presence in the online space. Businesses must also realize that only a properly qualified person/organization can ensure the performance of appropriate quality activities related to social media: in the 21$^{st}$ century, social media manager has become one of the most sought-after professions today.

The authors also sought the answer to why it is important for each business to appear on social media. The authors were surprised to find that in Slovakia, effective two-way communication with consumers was present among the three most significant factors, while in the case of Hungary, one-way communication was the focus. The authors recommend that businesses operating in Hungary place more emphasis on the implementation of two-way communication with consumers, for which individual social media platforms provide an excellent opportunity. Furthermore, effective communication with the target audience ensures the possibility of immediate feedback and the monitoring of rapidly changing consumer needs, thereby providing an opportunity to create a competitive advantage. During the third group of questions, the focus was on the follow-up of competing businesses on social media and the management of feedback from the target audience. Similar results were achieved in both countries: about two-thirds of the businesses follow the activities of their competitors online.

As was mentioned during the literature review, social media creates an excellent opportunity to monitor the activities of competing businesses, which can play an important role partly in a company's own development and partly in the development of a competitive advantage, the key foundational pillar of which is to understand competitors' strategies and continuous monitoring. In addition to monitoring, however, the authors consider it necessary to incorporate critical comments made towards competing businesses into organizational processes, as learning from mistakes made by competitors is an cost-effective solution, which is a critically important issue in the life of small and medium-

sized businesses. In this regard, the authors recommend to businesses operating in Slovakia that, in the event that they invest more time and energy in observing competing organizations, they should not ignore the critical comments made against them, the follow-up of which does not take more than time and energy expenditure, and strive for their organizational implementation.

In the fourth group of questions of the survey, the authors focused on the expansion of the presence in social media, as well as the examination of the company's mindset regarding its possible implementation methods. Before the question of expansion, however, the subject of marketing activities and marketing costs came to the fore. In both countries, the authors were surprised to find that the SME sector spends 0–25% of their activities and costs on online presence, the reason for which, in their opinion, consists of several parts. In part, since most of the surveyed businesses stated that they have been present in social media for a period of 1–5 years, they have relatively little experience. In addition, the fear of novelty and change can also be a determining factor. Related to this, although organizations have market indicators related to the user base of social media, they often do not see the potential inherent in social media. In this regard, the last influencing factor worth mentioning is the short-term organizational mindset. According to the authors' opinion, businesses often think along the lines that if the presence in social media does not fulfill the high hopes placed on it within a short period of time, in that case they are willing to bear such a large loss. As a result, the authors recommend that the companies of both countries prioritize possible profit instead of possible loss in their way of thinking. The results of the feedback received during the following question prove that during the period that has passed since the presence of small and medium-sized enterprises in social media, a positive change has taken place in the life of most organizations in both countries. In this regard, it can be stated that the time, energy and costs invested by the organizations will pay off, and businesses should invest in online presence. The part of the SME sector where companies feel that no changes have taken place since their presence in the online space, the authors recommend that they put more emphasis on monitoring their own target audience and, accordingly, think about making sure they use the right social media platform- e. In order for the positive effect to last, it is crucial that organizations do not stop following up with their target audience, as well as spin some of the revenue growth generated by the positive effect back into the social media presence. It can be stated as a critical observation that in the case of Hungary, there are a large number of businesses that do not plan to expand their presence in social media. Although this ratio is reversed in the case of Slovakia, the number of organizations that do not aim to expand their online presence is still high. Nowadays, a consumer no longer prefers just one social media platform, it is much more common to be present on several social media websites at the same time. For this reason, the authors recommend for the SME sector (focusing on the platforms preferred by their own and their potential target audience) to expand their presence in social media in order to create an opportunity to create two-way communication on a wider scale. In order for businesses to have the opportunity to spend more capital on their presence in social media in the future, the authors recommend that they return a certain percentage of the revenue increase realized as a result of the positive impact in the online space to the expansion of their presence in social media. Considering the practical way of implementation, it is further recommended that before making a selection, businesses should think about which expansion solution is the most popular and preferred for their own or their potential target audience. In this regard, the authors repeatedly highlight the importance of continuous monitoring of the target audience and the follow-up of competitors' strategies, since learning from the mistakes made by competing companies greatly facilitates faster adaptation. Today, the use of influencer marketing is extremely widespread, and it is preferred by small and medium-sized businesses as a result of its cost-effectiveness, but it is important to draw attention to the fact that this type of marketing activity does not generally produce the appropriate and expected results for all businesses. For this reason, it is necessary to prioritize the use of marketing activities tailored

to your own target audience. In the case of the last group of questions, the corporate success of the previously used traditional marketing tools came into focus. Compared with the previously asked question, which related to the extent to which presence in social media is important in the lives of the companies questioned, considering a five-point Likert scale for both countries, the authors realized a lower value, which allows conclude that in the life of small and medium-sized enterprises, the presence in social media plays a more important role in relation to marketing activities. In summary, based on the research results, the authors believe that the way of thinking of the interviewed small and medium-sized enterprises in the field of social media has recently greatly developed in a positive direction. However, the authors also recommend that, from the point of view of the marketing strategy, a more important role should be devoted to social media in order to ensure effective organizational functioning and presence in the media, and considering the possibility of expansion in the future, taking into account the suggestions made earlier. By accepting the critical observations, conclusions and suggestions made by the authors, and by incorporating them into their organizational system, businesses can greatly contribute to the development of the way of thinking related to social media in the small and medium-sized business sector.

Since the selected research topic is changing at an extremely fast pace thanks to the continuous development of information and communication technologies, in the opinion of the authors, it is worthwhile to carry out further research on the corporate application of social media on several fronts. For the future, we recommended case studies and expert interviews to reveal the relevant corporate needs, emerging obstacles and related solutions regarding social media. In addition, by using other research methods, it would be possible to ensure that the research questions reach the appropriate, competent persons. To imagine the possible future direction and development of corporate social media, it would be useful to repeat this research with more future-focused questions, and it would be worthwhile to expand the subjects of the research by translating the survey questionnaire into a third language, both for B2B and B2C organizations, which would further strengthen the statistical results. Another possible future direction is the extension of the research to consumers in order for the authors to provide more accurate answers for small and medium-sized enterprises regarding the preferences of consumers regarding the use of social media platforms and corporate marketing activities in the online space. in which direction and to what extent their respective needs change. After this has been implemented, it would be possible to deliver a document summarizing the research results and the practical implementation options formulated by the authors to the enterprises participating in the study. Furthermore, with regard to the hypotheses formulated by the authors, it would be advisable to define additional variables with the help of a research model, as well as to examine the possible relationships between them. In addition, it would be worthwhile in the future to place more emphasis on the study of the effects of the use of social media on organizational communication, trust, and corporate performance. Overall, corporate social media is an extremely exciting field of research that is constantly changing and requires further investigation to gain a deeper understanding of the issues related to the practical application and integration of social media.

**Author Contributions:** Conceptualization, E.K. and B.C.; methodology, E.K.; software, B.C.; validation, E.K.; formal analysis, B.C.; investigation, B.C.; resources, E.K.; data curation, E.K.; writing—original draft preparation, B.C.; writing—review and editing, E.K. and B.C.; visualization, B.C.; supervision, E.K. All authors have read and agreed to the published version of the manuscript.

**Funding:** This research was funded by PALLAS ATHÉNÉ DOMUS MERITI FOUNDATION.

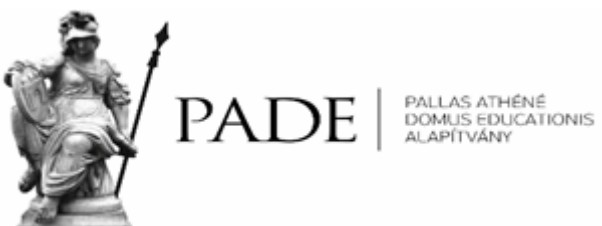

**Institutional Review Board Statement:** Not applicable.

**Informed Consent Statement:** Not applicable.

**Data Availability Statement:** Not applicable.

**Acknowledgments:** We would like to express our gratitude to the PALLAS ATHÉNÉ DOMUS MERITI FOUNDATION for supporting the successful implementation of our research

**Conflicts of Interest:** The authors declare no conflicts of interest.

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
