# Peer review of "Sustainable Competitiveness in the Case of SMEs—Opportunities Provided by Social Media in an International Comparison"

_sustainability, doi:10.3390/su141912505_

Round 1
Reviewer 1 Report
The findings presented are, in my opinion, conducted with sound research methods and tools. The hypotheses have been verified correctly. The proceedings have been discussed in a readable manner. 18 of the 45 literature items (45%) are from the last 5 years (2018 and newer). The results presented give a moderate level of utility to others in business practice which calls into question the value of the entire journal publication.
Author Response
Dear Reviewer!
As a first step, we would like to thank you for your constructive critical comments on our study, which we tried to incorporate into our article. We have supplemented our study with an introductory section, which provides assistance for a better overview. The field of corporate application of social media is extremely popular among researchers, however, we did not find studies dealing with the hypotheses formulated by the authors. As a result, we had little opportunity to coordinate our research results with previous empirical research in the conclusions chapter, which increases the significance of the research results. In addition to the two hypotheses formulated in the study, additional results are also presented, which, in the opinion of the authors, shed light on the mindset of small and medium-sized enterprises in the corporate integration of social media in the regions under investigation. The results highlight that the way of thinking of businesses has changed a lot about social media and more and more people think that social media is crucial from the corporate side, in Hungary and Slovakia there is still a high number of small and medium-sized businesses that do not attach much importance to social media. Taking into account the critical comments, in our future research we will strive to make the research results more useful for businesses in business practice.
Reviewer 2 Report
Dear Authors,
First of all I want to congratulate you for the efforts made along this research, which is quite interesting.
Although, I want to make some remarks regarding your references. I suggest you to use only those related references that were published in the last 1-3 years in journals indexed in WoS/CA or SCOPUS.
In the same time, please make efforts of research and put some more references to your paper from the papers already published in MDPI in the last 1-3 years.
Please go further with your research.
All the best !
Bye !
EG
Author Response
Reviewer 2
Dear Reviewer!
As a first step, we would like to thank you for congratulating us on our efforts during the research, as well as for the constructive critical comments, which can be used to further increase the quality of the study. The field of corporate application of social media is extremely popular among researchers, however, we did not find studies dealing with the hypotheses formulated by the authors. As a result, we had little opportunity to coordinate our research results with previous empirical research in the conclusions chapter, which increases the significance of the research results. Furthermore, taking into account the critical comments you suggested, we expanded the bibliography of our study with additional studies published in MDPI in the last 1-3 years, increasing the quality level of our study. Furthermore, having accepted the good advice, we will certainly try to continue our research in the future, and we will also strive to formulate further useful conclusions that will help small and medium-sized enterprises.
Reviewer 3 Report
Dear authors.
Below you will find my comments concerning you study.
Introduction section:
Line 38 and 40 start with same word. I advise you to change other word to use another word in one of the sentences.
The introduction section must be completely changed. It must be split in two sections: Introduction section and Literature review section.
Overall, the instruction section must be focused in portraying a clear idea of what you want to study and why. Furthermore, the main gap to be fulfilled must be provided. It is not the right place to make a literature review as you did.
After the introduction section, the following sections must be adapted to incorporate a literature review.
Section 2. Materials and Methods
Here you elaborate about the proposed hypotheses. However, without justify them. The right place to put them is the literature review, once it must be used to justify the development of the hypotheses.
Furthermore, this section must describe the procedures taken regarding to the measurement model. Justify why to use the model. Who developed the scale? Is it valid and reliable?
Section 3 Results is dependent of information that must be provided in section 2. You must indicate the data regarding to used scale.
Section 4 Discussion is weak. Results must be discussed and contrasted with what is known literature about the issue. However, you mostly discuss aspects regarding future research lines.
Section 5 partially presents information that could be relevant in the discussion section. I advise you to adapt both section 4 and 5.
Overall, I also believe that you should improve the section 5 Conclusion with a stronger focus on the link between the research objective and gap and how results fulfilled them.
Good luck with your work.
Author Response
Reviewer 3
Dear Reviewer!
As a first step, we would like to say a big thank you for the extremely detailed constructive critical comments, which can be used to greatly increase the quality of our study. Taking into account the primary critical comment you suggested, the introductory part has been completely revised, as this part has been divided into two chapters, an introduction part and a literature review part. As stated in the review, in the introductory part, we tried to introduce the study, and we put a lot of emphasis on giving the reader a comprehensive picture of exactly what and why we would like to research. After the introduction, a review of the literature related to the topic will be presented. In the second part of the study, based on the critical comments, we tried to improve and modify the materials and methods section by, among other things, presenting the measurement models in more detail, as well as explaining why we used these statistical methods to analyze the two formulated hypotheses. We tried to justify the presentation of the two hypotheses formulated, however, we did not find studies dealing with the two hypotheses formulated by us in the literature, which greatly complicated their justification in the literature review. The information corrected in the second section will help to review the third section. Moving on, and taking into account the critical comments, Sections 4 and 5 have been combined for better clarity, during which the first step is to present information that is relevant to the discussion section, after which the conclusions are drawn. At the end of the section, aspects related to future research directions were presented. The results highlight that the way of thinking of businesses has changed a lot about social media and more and more people think that social media is crucial from the corporate side, in Hungary and Slovakia there is still a high number of small and medium-sized businesses that do not attach much importance to social media. Taking into account the critical comments, in our future research we will strive to make the research results more useful for businesses in business practice.
Reviewer 4 Report
Thank you for the opportunity to read this paper. It is a interesting topic and I have some suggestion to improve the paper.
Abstract a better description of the methodology is indicated and maybe an argumentation why Hungary and Slovakia.
Introduction: What is the originality of this research? Paper research gap and originality should be better presented.
Maybe you can split in to parts a short introduction and the then description of the factors.
Methodology a better link to the existing literature is recommended. Why is this method the right one?
Results: a better argumentation and opinion is welcomed and a link to the existing literature and maybe highlighting what is new what is specific for Hungary and Slovakia.
Discussion also linked to the literature.
Conclusions to split in theoretical, managerial, and practical implications.
Additional recommendations:
Improving the literature.
Language review
Introduction improving the literature and the references – something new
Summary of Findings link to the existing literature
Good luck!
Author Response
Reviewer 4
Dear Reviewer!
As a first step, we would like to thank you for the constructive critical comments related to our study. In response to your critical comment, the Abstract was corrected, the methodology was explained more precisely, and an argument was also formulated, which related to why exactly Hungary and Slovakia were selected in terms of the research subjects. Furthermore, in the introduction chapter, which in the improved study is separated from the literature review for better clarity, we tried to highlight the importance of the research topic. The field of corporate application of social media is extremely popular among researchers, however, we did not find studies dealing with the hypotheses formulated by the authors. Regarding the methodological part, we tried to satisfy the critical comments, however, as mentioned earlier, we did not find any previous researches to examine the two hypotheses we formulated. Furthermore, in the second part of the study, based on the critical comments, we tried to improve and modify the materials and methods section by, among other things, presenting the measurement models in more detail, as well as explaining why we used these statistical methods to analyze the two hypotheses formulated. Taking into account the critical comments, the Discussion and Conclusion sections were slightly modified, as a result of which the two sections were combined for better transparency, during which the research results are presented as a first step, followed by the drawing of conclusions and the description of future research directions. As we mentioned earlier, since we did not find any previous research results to investigate the two hypotheses we formulated, it was a bit difficult for us to connect the existing results to the literature, however, taking into account the critical comments, we will strive to make the research results more useful in our future research. Accepting the additional recommendations, we expanded the bibliography of our study with additional studies published in MDPI in the last 1-3 years, increasing the quality level of our study. In addition, the language review was carried out and minor language errors were corrected.
Round 2
Reviewer 1 Report
I agree to accept in present form.
Reviewer 3 Report
Dear authors,
The changes made in the document improved its quality.
Good luck with your work.